# Terahertz Frequency-Modulated Continuous-Wave Inspection of an Ancient Enamel Plate

**DOI:** 10.3390/s25092928

**Published:** 2025-05-06

**Authors:** Frédéric Fauquet, Francesca Galluzzi, Rémy Chapoulie, Aurélie Mounier, Ayed Ben Amara, Patrick Mounaix

**Affiliations:** 1Institute from Material to System Laboratory, CNRS UMR 5218, Bordeaux INP, Université de. Bordeaux, F-33400 Talence, France; frederic.fauquet@ims-bordeaux.fr; 2Archéosciences Bordeaux—UMR 6034 CNRS, Université Bordeaux Montaigne, Domaine Universitaire, F-33607 Pessac, France; francesca.galluzzi@u-bordeaux-montaigne.fr (F.G.); chapouli@u-bordeaux-montaigne.fr (R.C.); aurelie.mounier@u-bordeaux-montaigne.fr (A.M.); abenamara@u-bordeaux-montaigne.fr (A.B.A.)

**Keywords:** millimeter wave radar imaging, terahertz imaging, thickness, fracture detection, non-destructive testing, ceramic, earthenware, polychrome glazes, pigment

## Abstract

This study investigates the application of terahertz frequency-modulated continuous-wave (FMCW) imaging for the non-destructive inspection of a historical enamel plate, using both reflection and transmission modes. A 300 GHz FMCW radar system was employed to capture high-resolution images of the plate’s internal and surface structures. Through optimized data acquisition and processing, the system successfully revealed subsurface features such as fractures, as well as surface-level textural variations linked to the decorative glazes. Although pigment differentiation remains a challenge, contrast variations observed in THz images suggest correlations with material composition. The results highlight the potential of FMCW terahertz imaging as a compact, rapid, and non-contact diagnostic tool for cultural heritage analysis. Its practicality and adaptability make it particularly suitable for in situ inspections in museums or restoration contexts.

## 1. Introduction

Terahertz radiation, which lies in the frequency range of 0.1 to 10 THz, has been applied to the study of cultural heritage objects for many years [1]. The most typical devices used for this purpose are pulsed terahertz time-domain systems [2,3], which allow for non-invasive cross-sectional imaging of the object by capturing the time it takes for the reflected THz pulses to return, and it can also be used for pigment identification [4]. Significant data related to the internal structure and stratigraphy of multilayered samples can be achieved using TDS [5,6], for example, a new method to accurately estimate the firing temperatures of ancient ceramic shards excavated from archeological sites [7] or the three-dimensional reconstruction of internal structure of stone-made objects [8]. With high axial resolution (minimum layer separation distance ~30–50 μm), TDS THz can be used to assess thin samples (~1 mm) [9,10]. Although the axial resolution of the FMCW method is only approximately 1–2 mm, it can be used to examine thicker samples (~5–15 cm) [11,12]. The THz FMCW radar can be categorized into two groups depending on the technological configuration, synthetic aperture and real-aperture imaging. Real-aperture imaging naturally provides stronger detection depth and more flexible detection options with respect to synthetic aperture imaging. Additionally, the antenna aperture and point spread function (PSF) limit the real-aperture radar [13]. Nevertheless, the THz FMCW are likely to increase their utilization into the conservation and restoration sectors than the time-domain systems because they can provide non-invasive surface and depth information about the scanned object, are faster (up to 10 kHz acquisition rate versus the ~100 Hz–1 kHz of the TD system), and are less expensive than pulsed terahertz systems. These systems have been used to analyze various heritage objects, such as paintings, and other ancient objects. FMCW terahertz imaging has emerged as a valuable tool for examining the internal structure of artwork. Ancient mummified samples (0.1 THz and 0.3 THz [14]) and easel pieces of art [15] have both been scanned using these radar systems. Additionally, details regarding internal characteristics are essential for assessing a glazed object′s conservation status or a ceramic object [16] since they can reveal detachments, inhomogeneities, or other flaws beneath the surface. This paper presents imaging techniques to optimize data extraction from FMCW THz scans of highly heterogeneous objects such as an old enamel plate. Additionally, we demonstrate that a 300 GHz FMCW radar can detect and quantify millimeter-scale fractures with a spatial resolution of up to 500 µm at a penetration depth of 0.7 cm. Our results highlight the potential of FMCW THz imaging for detecting small fractures and slight pigment variations in artwork due to reflectivity changes. The use of FMCW terahertz imaging in this study is motivated by its practical advantages over other THz techniques such as time-domain spectroscopy (TDS). Specifically, FMCW systems offer compactness, simplified setup, making them better suited for the in-situ diagnostics of fragile cultural heritage artifacts. While TDS provides richer spectral content, FMCW imaging offers a scalable, non-contact solution that can be more easily integrated into portable inspection tools. Available compact Silicon-based integrated radars make it particularly promising for heritage conservation where the ease of deployment and non-invasiveness are essential.

## 2. Materials and Methods

The process of creating vitreous enamel, also known as porcelain enamel, involves melting powdered glass onto a substrate at temperatures typically between 750 and 850 °C (1380 and 1560 °F). Once melted and flowed, the powder solidifies into a smooth, durable vitreous coating. The word “vitreous” comes from the Latin vitreus, meaning “glassy” or “like glass”. Enamel can be applied to any material that can withstand the high fusing temperatures, such as stone, glass, ceramic, or metal [17]. Technically, fired enamelware is a composite made of glass fused to another material or multiple layers of glass bonded together. Enameling is an ancient and widely used technique, historically associated with jewelry and decorative arts. Since the 18th century, enamel has also been applied to a variety of metal consumer goods, including cast-iron bathtubs, steel sinks, and certain cookware.

Longwy Pottery and Enamels is a historic company founded in 1798 in Longwy—then part of Moselle and now located in Meurthe-et-Moselle, France, near the borders of Belgium and Luxembourg. Over time, the original company has been succeeded by various entities, continuing its legacy. *Longwy enamels* are a distinctive form of glazed ceramics, crafted using a specialized technique and expertise recognized in France′s Inventory of Intangible Cultural Heritage. The presence of a monogram—specifically “AK” for Alfred Kirchtetter, known as the “enhancer”, or the artist responsible for applying pigments (mixed with frit) in delicate touches to the raised enamel—helps to date the piece to approximately the mid-twentieth century. His work adds illumination and richness to the decorative surface, enhancing its overall aesthetic appeal.

Our analyses were performed on a plate, manufactured by the Longwy Company, Longwy, France which is shown in Figure 1. The large thick layer of colored enamel creates a depth and volume of color that is typically impossible in ceramics that have been decorated. The plate is circular with a diameter of 45 cm and features a shallow concave profile, with the center positioned approximately 1 cm lower than the outer edge. This gradual biconvex curvature is typical of cast ceramics and contributes to the dynamic visual presentation of its decorative motifs.

At the center of the plate, a multicolored “phoenix” is depicted, surrounded by intricate ornamentation that extends outward. The design incorporates a rich palette of colored enamels applied in relief, a hallmark of Longwy craftsmanship. These enamels form slight convex cells due to the drop-by-drop application method, creating a textured surface with varied depth and light interaction.

The decoration includes bright pinks, dark greens, deep blues, and golden highlights, each corresponding to specific pigments or metallic inclusions. Some fine details, such as trees and leaf veins, are accented with highlighting strokes of enamel to enhance dimensionality.

The plate is made of white earthenware, and the enamel layers reach up to 650 µm in thickness, giving the motifs a vibrant, almost three-dimensional quality. The relief structure and enameling technique generate both color richness and depth, making the piece both technically sophisticated and artistically expressive.

Moreover, fractures and hairline cracks (crazing) were observed across the enamel layer, particularly near the phoenix figure and peripheral decorations. These are typical signs of aging due to internal tensions between the glaze and the biscuit substrate. The total thickness is about 7 mm. The little line that delineates the pattern and retains the colored glazes deposited in the various areas thus formed is the source of the name for relief glazes, as the name implies. The final painter used light touches of colors (pigments with frit) on the enamels in relief to add illumination and enhance the décor. Complementary details about the sample can be found in the study of [18].

## 3. Experiment

The plate was scanned using a 100 and 300 GHz system head in reflection mode at normal incidence using an FMCW imaging system (SynViewScan TRMF, manufactured by SynView GmbH, Bad Homburg, Germany). The existing THz FMCW radar mainly adopts a electronic frequency multiplier, which is widely employed in millimeter-wave radar systems due to its excellent high-speed switching capabilities and efficient performance at high frequencies. The frequency linearity is ensured by the Phase-Locked Loop (PLL) capability to follow the highly linear incremental digital ramp command sent by the Direct Digital Synthesizers (DDSs). A Voltage-Controlled Oscillator (VCO) introduced a strong non-linearity, which could drastically reduce the capabilities of an FMCW radar if not accounted for. With the correction applied by the PLL, the tuning voltage, VCO, then follows the inverse function of the VCO characteristic, over the operation frequency band [19]. According to Equations (1) and (2), the system′s achievable depth resolution is first restricted by the bandwidth of the frequency modulation (BW). Practically, a VCO frequency changes linearly between 0.23 THz and 0.32 THz to achieve an effective operating frequency of 300 GHz, with the BW equal to 0.09 THz. In the setup, Z is the direction of propagation of the radar beam. Z = 0 is the distance at the beam waist (in the middle of a pair of Teflon lenses), and Z > 0 is the distance closer to the transmission transceiver. The horizontal displacements (dX) and vertical (dY) step sizes for the 2-D raster scan were set to 0.5 mm. We reconstructed an image by selecting the maximum of the reflected or transmitted beam. Figure 2 is a schematic block diagram of the FMCW radar.

The reflected signal, delayed due to the propagation back and forth to the object, was guided toward the detection circuit, which includes a Low-Noise Amplifier (LNA), with a directional coupler linked to a multiplexer, where it is mixed with the reference signal or one of its lower harmonics. This mixing of the two beams will produce a beating signal, collected by a data acquisition unit (DAQ), at the frequency *f*_*m**i**x**e**r*_, directly proportional to the propagation delay, and, so, the distance to the object is presented in Equation (1) [19].(1)$$Z=\frac{c_{0}\Delta_{t}}{2n}=\frac{c_{0}f_{mixer}}{2n}.\frac{T_{sweep}}{BW}$$(2)$$dZ=\frac{c_{0}}{2nBW}$$
where $c_{0}$ is the speed of light in vacuum, *Δ*_*t*_ is the propagation-induced time delay, *B**W* is the accessible bandwidth, n is the refracting index of the propagation media at the working frequency, *f*_*m**i**x**e**r*_ is the output beating frequency of the mixer (1 MHz), and *T*_*s**w**e**e**p*_ is the frequency sweep period. From Equation (1), the longitudinal resolution can be derived as Equation (2) and mainly relies on the bandwidth of the radar transceiver.

The 100 GHz radar configuration provides enhanced penetration and is suitable for investigating subsurface structures, while the 300 GHz configuration offers superior lateral and longitudinal resolution due to its shorter wavelength. Depending on the material properties and imaging goals, users can select the optimal frequency for inspection. The high acquisition speed (several kHz) enables rapid non-destructive testing (NDT), with scan times typically lasting only a few minutes.

## 4. Results

### 4.1. Reflection: 100–300 GHz

The reflected images produced by the FMCW system are displayed in Figure 3a–c. The central bright region shown in Figure 3a and magnified in Figure 3c spans approximately 200 mm along the *X*-axis and 240 mm along the *Y*-axis, based on the figure′s axis scale. We collected the max reflectance for each pixel and used this value as parameter display. The phase shift between the reflected and received signals can be used by the FMCW system to select and display images with different times of flight. Since the depth resolution is only around 1 mm, we are unable to utilize this feature to establish a clear valid depth separation. In that example, this depth resolution is insufficient to discern the various layers of the ceramic. However, by measuring the waveform intensity for each pixel, we can extract qualitative features from the radar signal, such as interference patterns linked to general thickness variations and internal reflections.

The plate has a circular shape with fairly symmetrical radial patterns. The concentric rings in Figure 3a are likely related to the internal structure or surface texture of the ceramic. In radar imaging, the internal structure causes phase shifts, causing constructive and destructive interference patterns. The rings around the center may result from multiple interference phenomena with internal reflections, which are typical in high-frequency radar imaging. It is important to note that, to make ceramic objects by casting, industrials did not start from clay bread but from clay in liquid form. The piece of art was left to dry in a mold before being removed and air-dried. Then, ceramics were hand-painted on both sides, so the object was completely enabled. The structure is a tri-layer of enamel/clay/enamel (approximately 0.65 mm/4–5 mm/0.65 mm), which contributes to the observed interference patterns; however, the layer-specific resolution remains below the system′s longitudinal capability and thus cannot be individually resolved.

To qualitatively explain these features, a simple model can evaluate how constructive/destructive interference varies radially due to slight curvature or thickness variation, causing ring formation. The reflected intensity is equal to the following:I(λ) = I_0_ (1 + R^2^ + 2 R cos(4πnd/λ))(3)
where R is the reflection coefficient at each interface; n is the refractive index; d is the thickness of the layer; and λ is the THz wavelength.

To further support the hypothesis that the concentric rings observed in Figure 3a arise from internal reflections and interference phenomena, we provide a simple numerical analysis considering both the tri-layer structure (enamel–clay–enamel) and the geometry of the plate. The radar beam reflects between the internal dielectric interfaces, resulting in a variable phase shift due to local changes in optical path length.

Assuming the enamel layer has a thickness of d = 0.65 mm and a refractive index of n = 2.5 [18], the clay is several mm thick with an optical index of calcite around 1.43, and the basic optical path difference for a single round trip through the enamel is as follows:Δ = 2 n d cos (θ)(4)

At 100 GHz, Δ ≈ 3.25 mm, the free-space wavelength is λ = 3.0 mm, corresponding to a phase shift: ϕ = 2πΔ/λ = ≈ 6.8 rad.

This matches the 1.08th order of interference. However, the plate is not perfectly flat; the composition could also vary. It exhibits a slight conical shape with an inclination angle of approximately 10° from the edge toward the center. For normal-incidence imaging (i.e., the radar beam is perpendicular to the plate’s average surface), this geometry effectively increases the path length traversed by the wave in the enamel layer. The local apparent thickness slightly increases. This leads to an adjusted optical path difference: As the inclination changes radially, this leads to gradual radial modulation in phase, manifesting as concentric interference rings. The physical origin of these rings is therefore consistent with constructive and destructive interference arising from a multilayer system, much like a Fabry–Pérot cavity. We evaluate the radial distance between two destructive interference fringes. Taking into account the different thicknesses, a slope of 10°, and the different optical index, the radial distance is evaluated near 4 mm at 100 GHz, which is quite close to the fringe concentric ring distance in Figure 3a (around 4–5 mm). Small variations in thickness (±0.025 mm) cause the intensity to decrease significantly, showing how sensitive the interference is to enamel thickness.

At 100 GHz, one cannot discern clear details in the central part of the plate. The phoenix is barely detected in the reflected signal with the display parameter. Furthermore, other areas of different intensity can be observed, but they cannot be associated with a clearly different material used for the overall painting, except with the thin golden overlayer parts that present a higher reflectivity. We can also clearly perceive a big fracture on the left side (with a Lambda shape). The subsequent fractures are not detected with the selected display parameter. To distinguish the small fractures, we need to “travel all over “ with the help of the total video extracted from the radar measurements provided in Video 1. For this experiment, the plate is positioned at 50 mm from a Teflon lens.

Figure 3b shows the same ceramic plate measured by radar imaging at 300 GHz. Comparing it to the 100 GHz image, several observations and interpretations can be made. The 300 GHz image displays much finer details, particularly in the central region (see Figure 3c), revealing sophisticated patterns that were less distinct at 100 GHz. This is evidently due to the shorter wavelength at 300 GHz, which provides higher spatial resolution. Secondly, enhanced texture and surface features are found. These surface textures are visible, suggesting subtle variations in the ceramic′s surface roughness or microstructure (overlay painting of colors). The image analysis was significantly expanded to extract more meaningful information from the FMCW data. Although the depth resolution of the FMCW system (~1 mm) does not allow for precise quantification of the enamel layer thickness, spatial variations in signal intensity and interference patterns observed in the 300 GHz images provide indirect indications of structural heterogeneities. These include potential variations in local material composition, surface texture, or subtle internal inhomogeneities that are not accessible via optical inspection. Such features are likely to result from differences in the manufacturing process or variations in the firing conditions of the ceramic. The increased frequency enhances the sensitivity to small-scale permittivity variations, making these surface and internal features more prominent. For instance, reflectivity variation near the bird is observed with the same contrast observed using the TDS experiment [18]. The edge artifacts are more pronounced at 300 GHz, possibly due to increased diffraction or boundary reflections at higher frequencies. Also, horizontal lines in the background may be linked to system noise or scanning artifacts [20], which become more apparent with higher sensitivity. Again, the big Lambda shape fractures are easily detected, and its propagation can be followed by visualization of the 300 GHz reflected beam.

### 4.2. Transmission 100–300 GHz

Figure 4a shows the ceramic plate measured by radar imaging in transmission at 100 GHz. Comparing it to the reflection image at 100 GHz, several differences and observations can be made. The transmission image appears much more uniform with fewer interference patterns compared to the reflection mode. This is because transmitted waves are less affected by surface roughness and reflectivity variations. Higher contrast is observed around internal structures, and the propagation of the fractures inside the plate is detected. The dynamic range in dB is different from the reflection mode, highlighting different scattering mechanisms and attenuation paths. Also, transmission mode emphasizes absorption and phase shift through the material, whereas reflection emphasizes surface reflection and refraction. The dynamic range observed in the transmission images (Figure 4) is comparable to that in the reflection images (Figure 3). However, the smoother contrast in transmission mode is primarily due to different propagation mechanisms and attenuation effects, rather than a limitation in dynamic range. Based on the estimated dielectric parameters, the transmitted signal is expected to drop by nearly 95% over a 5 mm thickness, leading to significant signal attenuation and high contrast in transmission mode. We added a piece of metallic tape stuck on the back surface of the plate (a square and a polygon shape) to check the transparency of the multilayer ceramic object. Figure 4b presents the transmission-mode image of the enamel plate acquired at 300 GHz. Compared to the 100 GHz image (Figure 4a), this higher-frequency scan reveals significantly finer internal structures and micro-fractures within the ceramic material. Notably, the transmission image uncovers subtle internal features that remain invisible in optical or lower-frequency THz imaging, indicating potential variations in material density or firing inhomogeneities. These structural details are crucial for evaluating the conservation state of heritage ceramics. However, the image also shows attenuation artifacts, especially in the central area.

At last, a central darkest circular zone is visible for both frequencies. We also provided, in Figure 5a, the superposition of the visible and an X-ray image of the central part of the plate. Figure 5b is the rear visible image of the central zone of the plate, combined with a THz imaging scan performed with a time-domain spectro-imaging setup in reflection. So, a typical plate consists of several distinct parts. The well is the central, recessed area where food is served. The lip surrounds the well and is usually a flat or slightly inclined section that rises gently or runs parallel to the base. While often mistaken for the rim, the lip varies in width depending on the design and is not always present. The rim refers to the outermost edge of the plate, which is often decorated, for example, with gilding or painted details. The base (shown by a red dot circle in Figure 5) is the underside of the plate, sometimes featuring a foot ring to elevate it slightly from the surface that it rests on. In transmission mode, this added thickness of the base hides and alters the analysis in the central part of the plate of the surface details (see Figure 4 for FMCW transmission images and Figure 5a for X-ray image). The TDS THz image (inserted in the lower part) reveals internal structures of the plate that are invisible to the naked eye and on FMCW images. This is probably linked to variations in the reflected field and indicated differences in density during the firing of the ceramic, as well as during the casting of the piece. Plaster molds are generally used for the manufacture of pieces.

## 5. Discussion

The terahertz imaging performed at 100 GHz and 300 GHz on the ceramic plate highlights significant differences in penetration depth, spatial resolution, and defect contrast. At 100 GHz, the relatively large wavelength enables deeper penetration through the material, which is particularly advantageous for analyzing internal structures and identifying defects located at greater depths. On the other hand, imaging at 300 GHz provides significantly enhanced spatial resolution, allowing for better visualization of surface details and subtle structural heterogeneities. Due to the system’s limited axial resolution (~1 mm), it is not possible to directly resolve the enamel layer thickness. The observed variations in reflected intensity are therefore interpreted as relative modulations—likely resulting from interference patterns or changes in the local refractive index—rather than true cross-sectional enamel profiles. Interfaces, cracks, and density variations appear with higher contrast (see the video), facilitating the analysis of discontinuities within the material. However, this improved resolution comes at the cost of increased attenuation due to dielectric losses and scattering, thus reducing penetration depth. These results indicate that the choice of imaging frequency depends on the intended objective: 100 GHz imaging is more suitable for deep exploration and internal defect detection, whereas 300 GHz imaging provides a detailed mapping of surface structures and better detection of micro-defects even if the radar beam is efficient in transmission mode at 300 GHz. A combined approach using both frequencies could thus offer a more comprehensive analysis, leveraging both the penetration depth of 100 GHz and the precision of 300 GHz. Although this work proposes a joint interpretation of reflectance and transmission data at 100 and 300 GHz, experimental fusion of the two frequency bands remains a subject for future study. In particular, implementing pixel-level fusion or spectral co-registration would enable a more robust comparison between surface and subsurface features. Considering the maximal reflectivity image, once again, a pseudo-topographic map of the sample could be collected. From those preliminary visualization results on a complex shape sample, several very pragmatic problems are already addressed with dimensioning topics such as the detection and spatial estimation of cracks, or delaminations, as well as homogeneity inspection. Nonetheless, we can observe at 300 GHz a differential reflectivity [21] correlated to some overlayer painting such as the pink color on the trees, dark green on the leaves, and other colors on the phoenix’s throat and chest. The reflectivity maps at 300 GHz clearly reveal well-defined outlines that coincide with some of the colored motifs visible in the optical image, particularly areas painted with pink, green, and gold enamels. These spatially distinct zones suggest that terahertz imaging is sensitive to surface morphology and dielectric property variations. However, while these contrasts visually correspond to pigmented regions, they do not constitute direct evidence of pigment differentiation. No quantitative chemical analysis or pixel-level co-registration was performed in this study to formally link the THz reflectivity variations to material composition. Previous analyses using XRF and VNIR hyperspectral imaging [18] confirmed that certain areas are enriched with elements, such as Cu, Co, Cr, or Mn, but without synchronized comparison with the THz maps. Consequently, the observed contrasts are considered suggestive but not conclusive of dielectric variations related to pigments. Future work should combine co-registered THz imaging with chemical mapping techniques to rigorously assess the correlation between pigment composition and terahertz reflectivity. The THz signal is likely influenced by both material composition and surface morphology, making these observations complementary to prior chemical characterization. We have explicitly compared the results obtained via FMCW imaging with those from our previous terahertz time-domain spectroscopy (TDS) study on the same object [18]. The two approaches are thus not redundant but rather complementary, depending on the investigation goals and constraints. Video 2 and Video 3, which are superpositions of the transmitted and reflected images at 300 GHz and the visible one, allow to see this tiny detail of reflectivity changes due to the Fresnel coefficient between the top-most ceramic layer and air. Recent developments in FMCW radar signal processing, such as the three-dimensional reconstruction of layered structures described in the study of [22], demonstrate the growing potential of this technique for advanced material diagnostics. Although our current study does not yet implement full 3D reconstruction due to real sample geometry, such methods should be implemented in future work. The present study is intended as a demonstration of FMCW imaging feasibility for enamel objects, with a focus on operational accessibility and structural visualization. Compared to conventional time-domain terahertz spectroscopy (TDS) systems, which typically involve femtosecond lasers, optical delay stages, and bulky benchtop configurations, the FMCW system could be more compact and field-adaptable. With transceiver modules measuring approximately 120 mm × 60 mm × 70 mm (L/H/W) mm^3^ and weighing less than a kg [23], the setup is easily portable. These features allow for flexible deployment in non-laboratory environments such as museums, conservation sites, or archeological contexts.

## 6. Conclusions

The use of terahertz (THz) imaging for inspecting ceramics and art objects offers several promising perspectives due to its ability to provide non-destructive and non-contact analysis, reveal internal structures, and detect invisible defects. While TDS surpasses in fine structural analysis and hyperspectral imaging, FMCW stands out for its speed and ability to probe deeper structures, making both methods complementary depending on imaging requirements.

We demonstrated that millimeter wave FMCW imaging will become a useful tool in the field of art restoration and the ceramic community in the near future. The unique features of its non-contact and non-destructive nature and the correlation between allowing the observation of defects in both transmission and reflection modes strengthen the reliability of the imaging approach. One of the key advantages of the FMCW imaging system demonstrated in this study is its compactness and ease of deployment. The radar heads used (100/300 GHz) are housed in units measuring approximately 12 × 6 × 7 cm^3^ and weighing less than 1 kg each, excluding the data acquisition module. This stands in sharp contrast to typical time-domain terahertz (TDS) platforms, which often require large optical benches, femtosecond lasers, and precision delay lines—leading to systems weighing over 20 kg and occupying more than 1 m^2^ of laboratory space. In practical terms, this makes the FMCW system particularly well suited for in situ diagnostics in museums, archeological sites, and restoration environments, where space constraints and transportability are critical.

This study is based on a single representative historical ceramic sample, and, while the results demonstrate the potential of THz imaging, generalization across other materials and production periods remains to be confirmed. Furthermore, the intrinsic limitations of THz imaging—such as the compromise between resolution and penetration depth, and sensitivity to surface roughness—should be considered when interpreting reflectivity and transmission results. These constraints will guide the development of optimized imaging protocols in future studies. One direct application is the monitoring of the object degradation. THz systems could be used to track the evolution of cracks, delaminations, or alterations over time, offering a valuable tool for preventive conservation. A second possibility is about authentication and expertise analysis. Structural variations detected by THz imaging could help authenticate ancient ceramics and identify modern restorations that are invisible to other methods. The development of more compact and portable THz devices coupled with automation and artificial intelligence could be carried out. Integrating machine learning algorithms will probably enable the automatic analysis of THz images, and improving the detection of subtle defects and reducing interpretation time would facilitate on-site inspections in museums, restoration laboratories, and archeological sites.

## Figures and Tables

**Figure 1 sensors-25-02928-f001:**
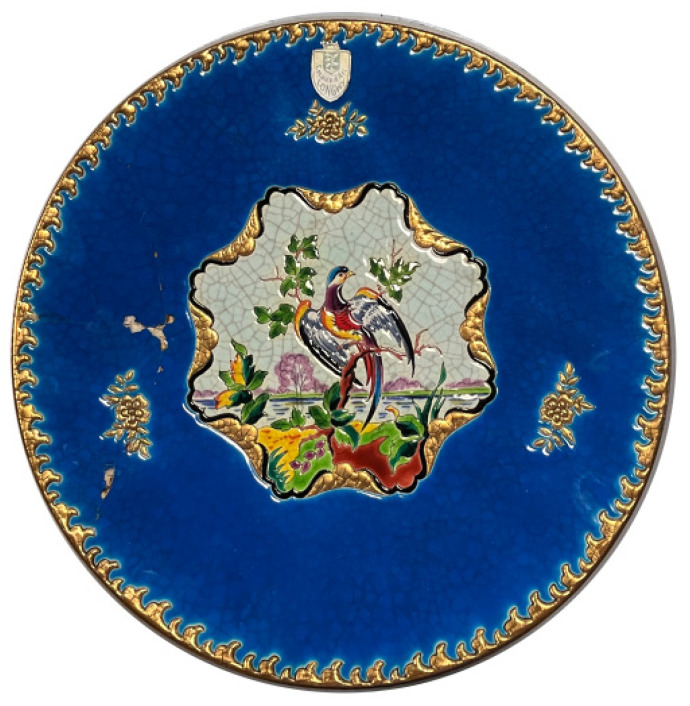
Overview of the enamel plate analyzed, showing its diameter (45 cm) and detailed design, including the phoenix motif at the center and surrounding decorative elements.

**Figure 2 sensors-25-02928-f002:**
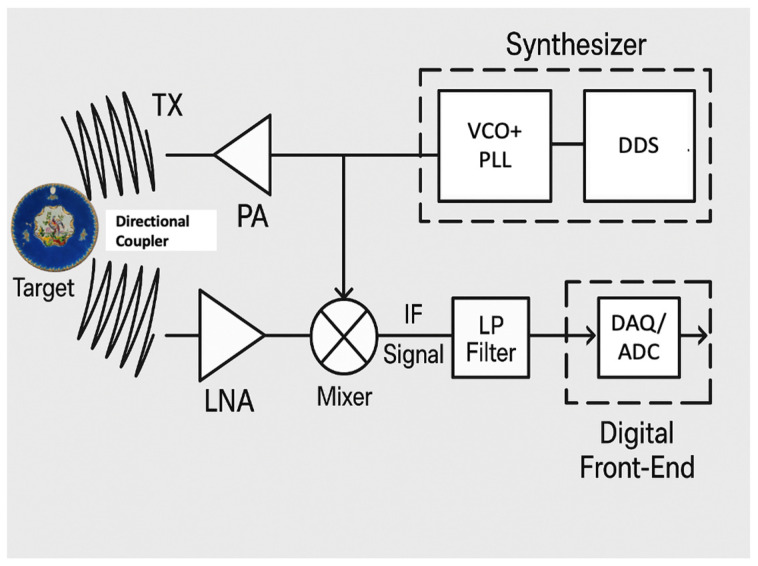
Functional block diagram of the frequency-modulated continuous-wave (FMCW) radar system used in this study. The Direct Digital Synthesizer (DDS) generates a highly linear digital ramp, which is used to drive a Phase-Locked Loop (PLL) that controls the Voltage-Controlled Oscillator (VCO). The output signal from the VCO is then amplified by a Power Amplifier (PA) before being transmitted through the TX antenna. The reflected signal, delayed due to the propagation back and forth to the object, is first amplified by a Low-Noise Amplifier (LNA) to minimize noise and then guided toward the detection circuit with a directional coupler linked to a multiplexer, where it is mixed with the reference signal. The resulting intermediate frequency (IF) beat signal is filtered and digitized by an Analog-to-Digital Converter (ADC), acting as the data acquisition (DAQ) system. The entire setup enables high-resolution distance measurements based on propagation-induced phase delays and beat frequency extraction.

**Figure 3 sensors-25-02928-f003:**
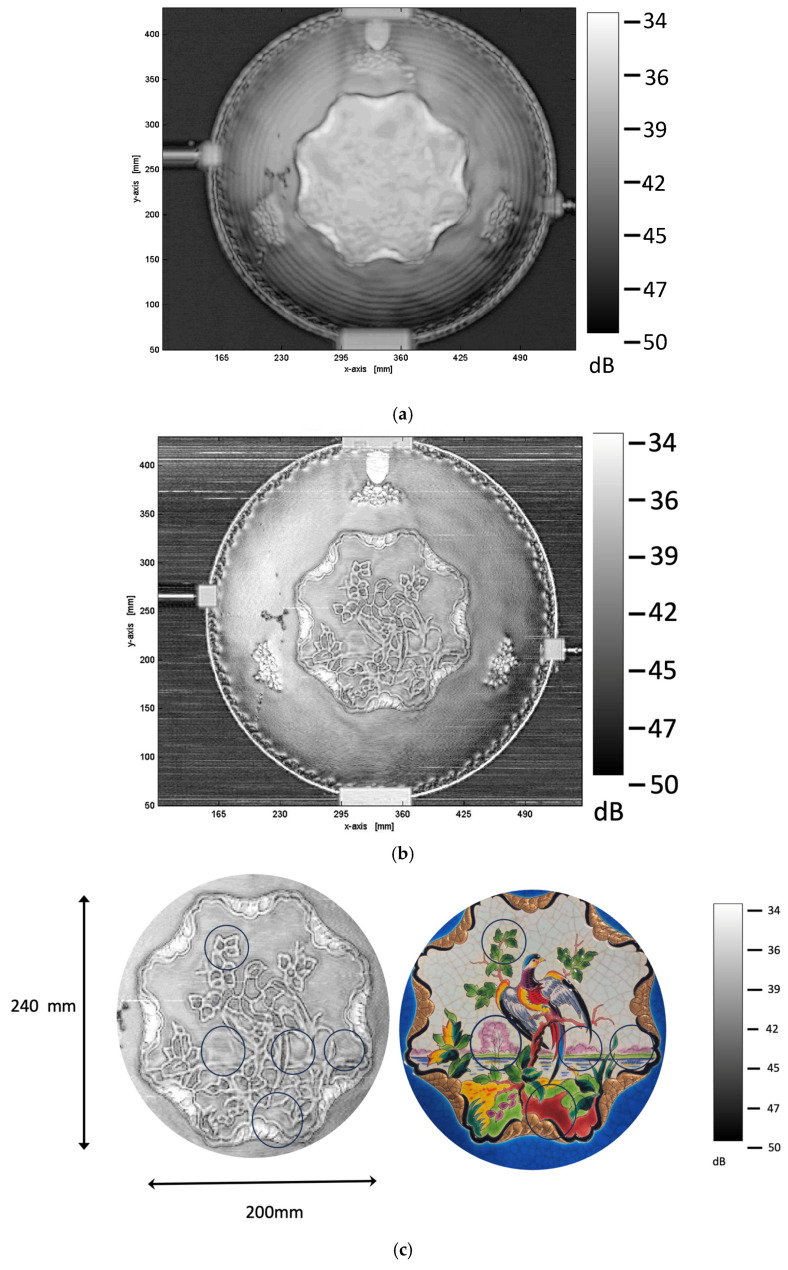
Reflected images obtained using the FMCW system at different frequencies: (**a**) 100 GHz reflection image, highlighting broad structural features and concentric patterns due to ceramic thickness variations. (**b**) The 300 GHz reflection image, revealing finer details of the artwork, including surface texture and microstructural inhomogeneities. (**c**) Zoom in the center of the plate. Significant reflectivity variations are observed, specifically in areas corresponding to pink, red, and dark green pigments. Some circular markings are linked to the visible light reference image (Figure 1), which serves as a guide to the decorative motifs and their colors.

**Figure 4 sensors-25-02928-f004:**
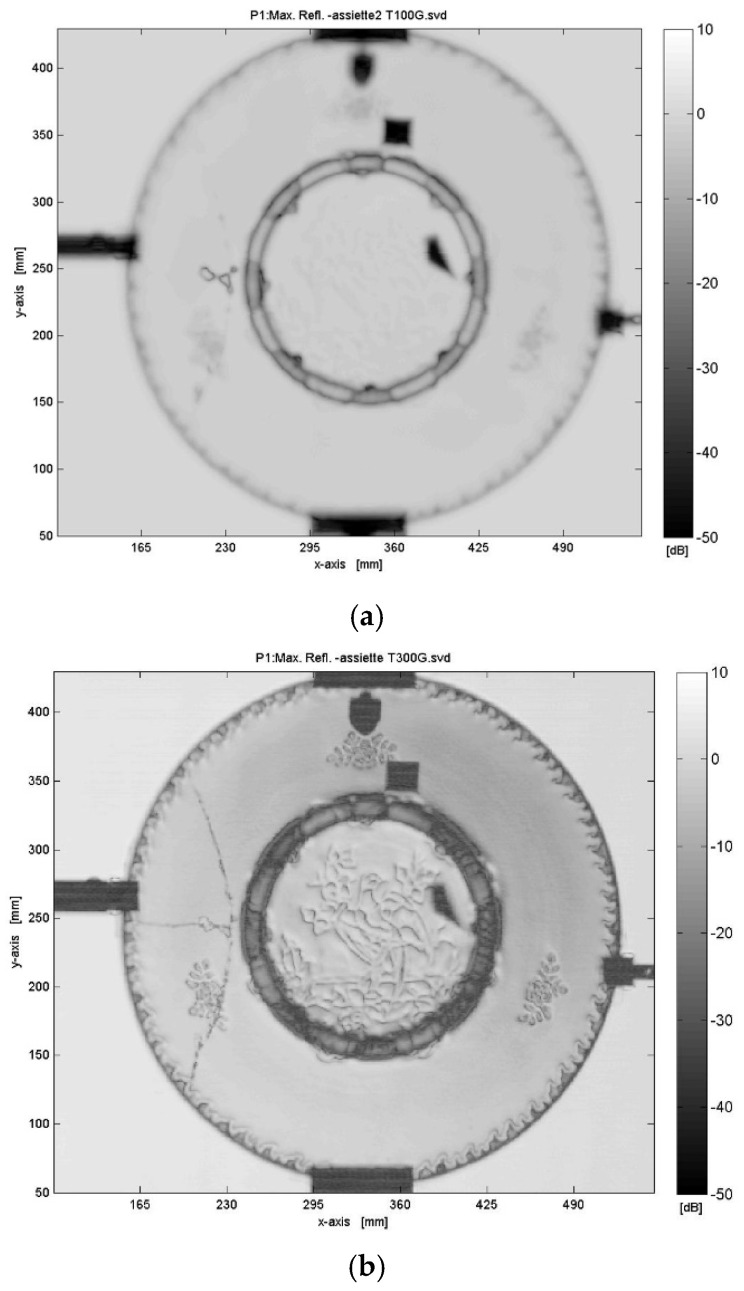
Transmission imaging results at different frequencies: (**a**) 100 GHz transmission image, displaying deep penetration with lower resolution and smoother contrast and (**b**) 300 GHz transmission image, providing higher resolution and revealing finer internal structures and micro-fractures.

**Figure 5 sensors-25-02928-f005:**
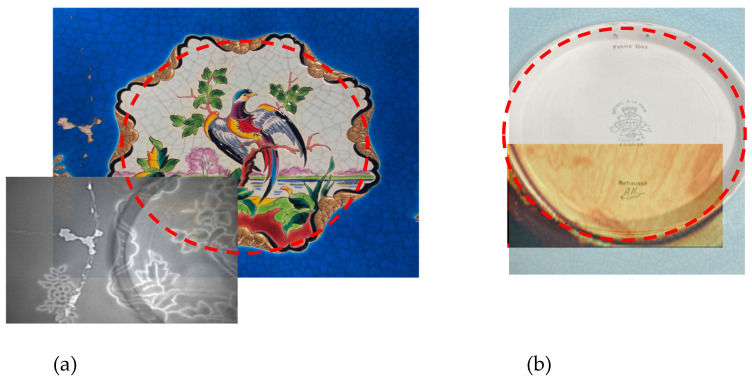
(**a**) visible image and X-ray of the center of the plate, (**b**) Rear view of the enamel plate, combined with a THz imaging scan using a time-domain spectroscopy (TDS) system in reflection mode.

## Data Availability

Data used are available upon request.

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
