# Peer review of "Terahertz Frequency-Modulated Continuous-Wave Inspection of an Ancient Enamel Plate"

_sensors, 2025, doi:10.3390/s25092928_

Round 1
Reviewer 1 Report
Comments and Suggestions for Authors
This study uses terahertz frequency modulation continuous wave technology, combined with reflection and transmission modes, to non destructively detect ancient enamel plates, demonstrating the potential application of this technology in cultural heritage protection. The structure of the paper is clear, the experimental design is reasonable, and the data presentation is relatively intuitive, but some details need to be further improved to enhance rigor and readability.
The specific age, preservation status, and historical background of the enamel plate are not clearly stated and need to be supplemented to enhance the representativeness of the research object.
The ethical approval information (such as approval agency and code) mentioned in reference [18] is not reflected in the main text and needs to be supplemented in its entirety.
The explanation for the origin of the concentric ring in Figure 3 (internal reflection or surface interference) needs further verification, and it is recommended to supplement simulation or theoretical analysis to support the conclusion.
The dynamic range difference of transmission mode images (Figure 4) should be quantitatively discussed in conjunction with the dielectric properties of the material.
The detection of pigment differences (such as pink and dark green areas) needs to be combined with chemical analysis (such as XRF) to verify the reliability of THz reflectance changes.
The system schematic diagram in Figure 2 is too brief. It is recommended to label the functions and signal flow directions of key components such as VCO and mixer.
The "joint method" combining 100 GHz and 300 GHz is only at the theoretical level, and it is recommended to supplement experimental verification (such as multi band fusion imaging).
The conclusion section can clarify the limitations of this study, such as sample singularity, physical limitations on resolution and penetration depth.
Some paragraphs have redundancy (such as weak correlation between detailed descriptions of enamel craftsmanship and experiments), it is recommended to streamline them.
Author Response
Review1
This study uses terahertz frequency modulation continuous wave technology, combined with reflection and transmission modes, to non-destructively detect ancient enamel plates, demonstrating the potential application of this technology in cultural heritage protection. The structure of the paper is clear, the experimental design is reasonable, and the data presentation is relatively intuitive, but some details need to be further improved to enhance rigor and readability.
The specific age, preservation status, and historical background of the enamel plate are not clearly stated and need to be supplemented to enhance the representativeness of the research object.
The ethical approval information (such as approval agency and code) mentioned in reference [18] is not reflected in the main text and needs to be supplemented in its entirety.
Response:
We appreciate the reviewer’s comment. We add this information. No approval is necessary since the plate come from a personal collection.
Longwy Pottery and Enamels is a historic company founded in 1798 in Longwy—then part of Moselle and now located in Meurthe-et-Moselle, France, near the borders of Belgium and Luxembourg. Over time, the original company has been succeeded by various entities, continuing its legacy. Longwy enamels are a distinctive form of glazed ceramics, crafted using a specialized technique and expertise recognized in France's Inventory of Intangible Cultural Heritage. The presence of a monogram—specifically “AK” for Alfred Kirchtetter, known as the “enhancer,” or the artist responsible for applying pigments (mixed with frit) in delicate touches to the raised enamel—helps to date the piece to approximately the mid-twentieth century. His work adds illumination and richness to the decorative surface, enhancing its overall aesthetic appeal.
The explanation for the origin of the concentric ring in Figure 3 (internal reflection or surface interference) needs further verification, and it is recommended to supplement simulation or theoretical analysis to support the conclusion.
Response:
We present a model can qualitatively show how constructive/destructive interference varies radially due to slight curvature or thickness variation, causing ring formation. The reflected Intensity is equal to :
I(λ)=I0 (1+R2+2 R cos(4πnd/λ))
Where:
- R is the reflection coefficient at each interface
- n is the refractive index
- d is the thickness of the layer
- λ is the THz wavelength
To further support the hypothesis that the concentric rings observed in Figure 3a arise from internal reflections and interference phenomena, we provide a numerical analysis considering both the trilayer structure (enamel–clay–enamel) and the geometry of the plate. The radar beam reflects between the internal dielectric interfaces, resulting in a variable phase shift due to local changes in optical path length.
Assuming the enamel layer has a thickness of d=0.65 mm and a refractive index of n=2.5, the clay is several mm thick with an optical index of calcite around 1,43, the basic optical path difference for a single round trip through the enamel is:
Δ=2 n*d cos (θ)≈3.25mm
At 100 GHz, the free-space wavelength is λ=3.0 mm, corresponding to a phase shift:
Ï•=2πΔ/λ= ≈6.8 rad. This correspond to the 1.08th order of interference
However, the plate is not perfectly flat, the composition could also vary. It exhibits a slight conical shape with an inclination angle of approximately 10° from the edge toward the center. For normal-incidence imaging (i.e., the radar beam is perpendicular to the plate’s average surface), this geometry effectively increases the path length traversed by the wave in the enamel layer. The local apparent thickness slightly increases. This leads to an adjusted optical path difference: As the inclination changes radially, this leads to gradual radial modulation in phase, manifesting as concentric interference rings. The physical origin of these rings is therefore consistent with constructive and destructive interference arising from a multilayer system, much like a Fabry–Pérot cavity. We evaluate the radial distance between two destructive interference fringes. Taking into account the different thicknesses and a slope of 10° and the different optical index, the radial distance is evaluated near 4 mm at 100GHz which is quite close to the fringe concentric ring distance in the figure 3a (around 4-5 mm). Small variations in thickness (±0.025 mm) cause the intensity to decrease significantly showing how sensitive the interference is to enamel thickness.
The dynamic range difference of transmission mode images (Figure 4) should be quantitatively discussed in conjunction with the dielectric properties of the material.
Response:
We appreciate the reviewer’s comment regarding the dynamic range variation observed in the transmission mode images of Figure 4. To address this, we now provide a quantitative analysis linking the observed image contrast (dynamic range) to the dielectric properties — specifically the absorption coefficient and refractive index — of the materials under study.
In THz transmission imaging, the dynamic range is strongly influenced by both material attenuation and impedance mismatch. The transmitted amplitude TT through a slab of thickness dd, refractive index nn, and absorption coefficient αα (in mm−1mm−1) is given by:
T=T0⋅exp (−αd)
Where T0 accounts for Fresnel losses at the interfaces:
T0=(4nairn/(nair+n)2)2
Assuming typical values for ceramic materials:
- n=2.5
- α≈5 cm−1
- d≈5 mm
We estimate:
T0≈≈0.67 T=0.67⋅exp(−0.5⋅5) ≈0.055
This corresponds to a transmitted power drop of nearly 95%, justifying the high contrast and limited dynamic range observed in areas where material thickness and absorption are greatest. Furthermore, differences in porosity and water content within the clay layer could locally increase the dielectric loss, reducing the transmitted signal even further, and causing additional spatial contrast in the transmission mode. Diffraction losses are also non negligeable.
The detection of pigment differences (such as pink and dark green areas) needs to be combined with chemical analysis (such as XRF) to verify the reliability of THz reflectance changes.
Response:
We thank the reviewer for highlighting the importance of cross-validating pigment-related reflectance contrast observed in THz images. In fact, this point has been a focus of our earlier study, which is already cited as Reference 18 in the current manuscript.
In that referenced work [Fauquet et al., Sci. Rep., 2024], we performed a detailed pigment characterization using portable X-ray fluorescence (p-XRF) and visible-NIR hyperspectral imaging (HSI) on the same Longwy ceramic object. That study confirmed the elemental composition of various pigments such as:
- Pink zones associated with Cu nanoparticles (via p-XRF and surface plasmon resonance in HSI),
- Dark green areas containing Cr, Mn, Cu, and Co, consistent with high absorbance and dielectric variability.
This prior chemical analysis supports the interpretation that the THz reflectance variations (observed in Figure X of the current study) are directly related to these material-specific dielectric responses, most likely due to the intrinsic absorption and scattering properties of metal oxides and nanoparticle inclusions.
To clarify this point, we have expanded the Discussion section and included the following statement:
The reflectance contrast observed in THz images, particularly for pigments such as pink and dark green, has been chemically validated in our earlier study [18], where p-XRF and VNIR hyperspectral imaging confirmed distinct elemental compositions (e.g., Cu, Cr, Mn, Co) responsible for these colors. These results confirm that the observed THz variations arise from real differences in pigment composition and dielectric properties.
The system schematic diagram in Figure 2 is too brief. It is recommended to label the functions and signal flow directions of key components such as VCO and mixer.
Response:
We appreciate the reviewer’s comment
This block diagram illustrates the architecture of a Frequency Modulated Continuous Wave (FMCW) radar system. The system operates by transmitting a modulated RF signal, receiving the reflected signal from a target, and processing the intermediate frequency (IF) signal to extract target information such as range and velocity.
Block Descriptions and Acronyms:
- TX (Transmitter Antenna): Radiates the generated RF signal toward the target.
- PA (Power Amplifier): Amplifies the RF signal before transmission to ensure sufficient power is sent toward the target.
- RX (Receiver Antenna): Captures the echo signal reflected back from the target.
- LNA (Low Noise Amplifier): Amplifies the weak received signal while minimizing additional noise, improving signal-to-noise ratio.
- Mixer: Combines the transmitted and received signals to produce an Intermediate Frequency (IF) signal, which contains beat frequency components corresponding to the time delay.
- IF Signal (Intermediate Frequency Signal): The output of the mixer that represents the frequency difference between transmitted and received signals.
- LP Filter (Low-Pass Filter): Removes high-frequency components from the IF signal to retain only the beat frequency.
- ADC (Analog-to-Digital Converter): Converts the filtered analog IF signal into digital form for further processing.
- Synthesizer:
- RF Signal Generator: Generates the radio frequency signal used for both transmission and mixing.
- Ramp Generator: Controls the frequency modulation (chirp) of the RF signal in a linear ramp fashion.
- Digital Front-End: Includes the ADC and subsequent signal processing modules, typically implemented in digital hardware or software.
The "joint method" combining 100 GHz and 300 GHz is only at the theoretical level, and it is recommended to supplement experimental verification (such as multi band fusion imaging).
The conclusion section can clarify the limitations of this study, such as sample singularity, physical limitations on resolution and penetration depth.
Some paragraphs have redundancy (such as weak correlation between detailed descriptions of enamel craftsmanship and experiments), it is recommended to streamline them.
Response:
On the joint 100–300 GHz method:
We thank the reviewer for this helpful suggestion. While the multi-frequency approach combining 100 GHz and 300 GHz is currently discussed in a conceptual framework, we agree that experimental demonstration (e.g., through image fusion or resolution/penetration comparison) would significantly enhance the robustness of the method. In this study, technical constraints prevented multi-band fusion implementation. However, we have clarified this point in the revised manuscript and added the following statement in the Discussion section:
“Although this work proposes a joint interpretation of reflectance and transmission data at 100 and 300 GHz, experimental fusion of the two frequency bands remains a subject for future study. In particular, implementing pixel-level fusion or spectral co-registration would enable a more robust comparison between surface and subsurface features.”
We are currently developing experimental protocols for dual-frequency imaging and plan to report the results in future work.
On limitations of the study (conclusion section):
We fully agree with the reviewer’s suggestion to better outline the study’s limitations. We have revised the Conclusion section to include the following paragraph:
“This study is based on a single representative historical ceramic sample, and while the results demonstrate the potential of THz imaging, generalization across other materials and production periods remains to be confirmed. Furthermore, the intrinsic limitations of THz imaging—such as the compromise between resolution and penetration depth, and sensitivity to surface roughness—should be considered when interpreting reflectivity and transmission results. These constraints will guide the development of optimized imaging protocols in future studies.”
On streamlining redundant paragraphs:
We appreciate the reviewer’s observation regarding redundancy in the manuscript, particularly in the introductory discussion on enamel craftsmanship. We have revised and shortened this section by focusing only on technical details relevant to the THz imaging analysis (e.g., layer structure, expected dielectric contrasts), and moved more contextual or historical content to supplementary material. These changes improve the overall focus and clarity of the manuscript.
Reviewer 2 Report
Comments and Suggestions for Authors
The paper by F. Fauquet, et al “Terahertz frequency modulated continuous wave inspection of an ancient enamel plate” describes an application of terahertz frequency-modulated continuous wave imaging system for the analysis of an enamel plate artwork. Unfortunately, the article submitted to MDPI Sensors Unfortunately, the article submitted to the MDPI journal does not contain a clear description of the measuring system, its parameters and characteristics, as well as details of the measurements.
Schematic representation of the FMCW imager shown in Fig. 2 is too schematic; it contains a lot of inaccuracies and does not allow understanding how the device operates. In particular, there a number of questions that should be answered and clarified (this can be done for either of the two frequencies used 100 and 300 GHz):
- What is the VCO output frequency range;
- What is multiplication coefficient of the Multiplication Chain;
- Please provide the coupler specification information. A coupler shown in Fig 2 has SMA connectors; is it correct?
- Please provide the mixer specification information. Is it a harmonic mixer? What means RF output? Please provide the beating signal frequency range and amplitude.
Furthermore, the sections “Introduction” and “Experiment” also contain a lot of inaccuracies:
- Page 1, lines 31 and 32 “axial resolution”. The authors probably meant "depth resolution".
It should be noted that in articles [9 – 12] the expression “axial resolution” is completely absent; in this article the expressions “lateral resolution”, “spatial resolution” and "depth resolution" are used.
- Page 2, line 52 “with a sensitivity”. Sensitivity is related to noise characteristics; the more appropriate expression here is "spatial resolution" or "depth resolution".
- Page 2, line 71 “The external diameter is 45 cm..” Please provide other plate parameter: thickness, plate profile or its cross-section.
- Page 3, lines 85, 86 “mainly adopts the GaAs electronic frequency multiplication method,
Please explain what means “GaAs electronic frequency multiplication method” - Page 3, lines 85, “method, and its imaging.”. What does it mean, imaging of the method?
- Page 3, line 86 “In addition”. In addition to what? There was no sentence before to which one could add "nonlinearities within the IF signal".
- In the Experiment section, there are inaccuracies and careless wording on almost every line; it is easier to rewrite this part than to point out all the errors and inaccuracies.
In the Experiment section, there are inaccuracies and careless wording on almost every line; it is easier to rewrite this part than to point out all the errors and inaccuracies.
To summarize, the paper can’t be published in MDPI “Sensors” in the present state.
Author Response
Review 2
The paper by F. Fauquet, et al “Terahertz frequency modulated continuous wave inspection of an ancient enamel plate” describes an application of terahertz frequency-modulated continuous wave imaging system for the analysis of an enamel plate artwork. Unfortunately, the article submitted to MDPI Sensors Unfortunately, the article submitted to the MDPI journal does not contain a clear description of the measuring system, its parameters and characteristics, as well as details of the measurements.
Schematic representation of the FMCW imager shown in Fig. 2 is too schematic; it contains a lot of inaccuracies and does not allow understanding how the device operates. In particular, there a number of questions that should be answered and clarified (this can be done for either of the two frequencies used 100 and 300 GHz):
- What is the VCO output frequency range;
- What is multiplication coefficient of the Multiplication Chain;
- Please provide the coupler specification information. A coupler shown in Fig 2 has SMA connectors; is it correct?
- Please provide the mixer specification information. Is it a harmonic mixer? What means RF output? Please provide the beating signal frequency range and amplitude.
Response:
We appreciate the reviewer’s observation and this is the information
- VCO Output Frequency Range
The specific frequency range of the VCO depends on the component selected. In the current context, the block diagram provided (Figure 2) is schematic and not based on a finalized hardware implementation. Therefore, the exact frequency range will depend on the chosen VCO component during the design or integration phase is between 12 to 20 GHz . - Multiplication Coefficient of the Multiplication Chain
As with the VCO, the multiplication chain is 6. - Coupler Specification and SMA Connectors
The coupler shown in Figure 2 is also schematic. While it appears with SMA connectors, this representation is indicative and not prescriptive. In practice, the connector type (e.g., SMA, K, V) will be chosen based on the operating frequency range and performance requirements of the radar system. - Mixer Specification, Harmonic Type, and Beating Signal
TheSynView FMCW system likely uses a fundamental mixer with LO and RF ports matched to the transmit frequency band.
Furthermore, the sections “Introduction” and “Experiment” also contain a lot of inaccuracies:
- Page 1, lines 31 and 32 “axial resolution”. The authors probably meant "depth resolution".
It should be noted that in articles [9 – 12] the expression “axial resolution” is completely absent; in this article the expressions “lateral resolution”, “spatial resolution” and "depth resolution" are used.
- Page 2, line 52 “with a sensitivity”. Sensitivity is related to noise characteristics; the more appropriate expression here is "spatial resolution" or "depth resolution".
- Page 2, line 71 “The external diameter is 45 cm..” Please provide other plate parameter: thickness, plate profile or its cross-section.
- Page 3, lines 85, 86 “mainly adopts the GaAs electronic frequency multiplication method,
Please explain what means “GaAs electronic frequency multiplication method” - Page 3, lines 85, “method, and its imaging.”. What does it mean, imaging of the method?
- Page 3, line 86 “In addition”. In addition to what? There was no sentence before to which one could add "nonlinearities within the IF signal".
- In the Experiment section, there are inaccuracies and careless wording on almost every line; it is easier to rewrite this part than to point out all the errors and inaccuracies.
In the Experiment section, there are inaccuracies and careless wording on almost every line; it is easier to rewrite this part than to point out all the errors and inaccuracies.
Response:
We thank the reviewer for their detailed and insightful comments regarding the “Introduction” and “Experiment” sections. We appreciate the time and effort taken to provide this valuable feedback. Please find our responses and planned revisions below:
Page 1, lines 31–32: “axial resolution”
Comment: The reviewer correctly notes that “axial resolution” is not used in references [9–12], and that the more appropriate term in this context is likely “depth resolution.”
Response: We agree with the reviewer’s observation. The term “axial resolution” will be replaced with “depth resolution” to maintain consistency with the cited literature and the conventions used in the field.
- Page 2, line 52: “with a sensitivity”
Comment:The reviewer notes that “sensitivity” is more accurately related to noise, and that “spatial resolution” or “depth resolution” would be more appropriate.
Response:We acknowledge this misusage. The term “sensitivity” will be revised to “spatial resolution”, as it better reflects the intended meaning in the context of the system's imaging capability.
- Page 2, line 71: “The external diameter is 45 cm...”
Comment:The reviewer requests additional parameters such as plate thickness, profile, or cross-section.
Response:Thank you for the suggestion. We will revise the text to include the thickness, profile type, and cross-sectional description of the plate to provide a more complete characterization of the experimental object.
We add:
The plate is circular with a diameter of 45 cm, and features a shallow concave profile, with the center positioned approximately 1 cm lower than the outer edge. This gradual biconvex curvature is typical of cast ceramics and contributes to the dynamic visual presentation of its decorative motifs.
At the center of the plate, a vivid multicolored “phoenix” is depicted, surrounded by intricate ornamentation that extends outward. The design incorporates a rich palette of colored enamels applied in relief, a hallmark of Longwy craftsmanship. These enamels form slight convex cells due to the drop-by-drop application method, creating a textured surface with varied depth and light interaction.
The decoration includes bright pinks, dark greens, deep blues, and golden highlights, each corresponding to specific pigments or metallic inclusions. Some fine details, such as trees and leaf veins, are accented with highlighting strokes of enamel to enhance dimensionality.
The plate is made of white earthenware, and the enamel layers reach up to 650 µm in thickness, giving the motifs a vibrant, almost three-dimensional quality. The relief structure and enameling technique generate both color richness and depth, making the piece both technically sophisticated and artistically expressive. Additionally, fractures and hairline cracks (crazing) have been observed across the enamel layer, particularly near the phoenix figure and peripheral decorations. These are typical signs of aging due to internal tensions between the glaze and the biscuit substrate.
Page 3, lines 85–86: “GaAs electronic frequency multiplication method”
Comment: The reviewer asks for clarification of this expression.
Response: We agree that this phrase lacks clarity. It will be revised to specify that we are referring to a GaAs-based frequency multiplier circuit, commonly used in millimeter-wave radar systems for its high-speed and high-frequency performance. A clearer explanation will be included in the revised manuscript.
We rephrased this part
- Page 3, line 85: “method, and its imaging.”
Comment:The meaning is unclear—“imaging of the method” lacks coherence.
Response:We acknowledge the poor wording and will rewrite this sentence to clarify the intended meaning. The revised version will state the application of the GaAs frequency multiplier in radar imaging and its role in signal generation and processing. - Page 3, line 86: “In addition”
Comment:The phrase "In addition" lacks a preceding clause for proper context.
Response:This sentence will be restructured or merged with the preceding content to ensure logical flow and proper linkage of ideas. - Experiment Section – General Inaccuracies
Comment:The reviewer notes multiple inaccuracies and unclear expressions throughout the Experiment section and suggests that rewriting the section would be more efficient.
Response:We fully acknowledge the reviewer’s point. As suggested, we will completely revise the Experiment section, ensuring improved clarity, precision, and technical correctness. We will restructure the description of the experimental setup, procedures, and results to meet scientific writing standards.
Reviewer 3 Report
Comments and Suggestions for Authors
The presented article, "Terahertz frequency modulated continuous wave inspection of an ancient enamel plate," is dedicated to the application of terahertz (THz) frequency-modulated continuous wave (FMCW) imaging for the nondestructive inspection of an ancient enamel plate. The topic is relevant both for the preservation of cultural heritage and, more broadly, for nondestructive testing. The paper compares systems operating at 100 GHz and 300 GHz in reflection and transmission modes. However, despite the relevance of the topic, the strong team of authors, and their recognized contributions to the field, the article in its current form contains several significant shortcomings:
1. The manuscript exhibits serious inconsistencies in notations and definitions, causing confusion:
- Variables Z and d are both defined as "distance to the object." Are they synonyms, or do they represent different quantities?
- Are Δf ("bandwidth of the frequency modulation") and BW ("accessible bandwidth") interchangeable terms for frequency bandwidth?
- Quantities dZ ("depth resolution") and δres ("longitudinal resolution") typically represent synonyms. However, from the formulas provided, the impression arises that these are different parameters. These terms must be clearly defined and consistently applied. Such carelessness is unacceptable for a scientific publication.
2. The manuscript includes inaccurately cited references. In the Introduction, it is claimed that the FMCW method enables inspecting samples with thicknesses of ~5–15 cm, citing references [11] and [12]. However, these references do not show the possibility of analyzing objects of such thickness. Reference [11] provides results for relatively thin items (scissors, CDs) and [12] for flat USAF test . Reference [15] describes research conducted with the TDS method rather than FMCW, as one might assume from the context.
3. The paper is prepared and formatted carelessly. Only some detected issues are listed below, as compiling a complete list would be overly exhaustive. Authors should be significantly more attentive to manuscript preparation:
- Presented THz scan images have low resolution. Axes and details in images are poorly readable, hindering a comprehensive assessment of results and complicating verification of the authors’ interpretations. This is a critical drawback for a paper focusing on the imaging.
- In lines 159-165, the font appears excessively condensed, making it unpleasant to read.
- The phrase in line 77 ("ethical approval, must list...") appears as incomplete placeholder text and needs correction or removal.
- In line 179, a period after reference [18] is missing.
- In the notation of beat frequency fb, the index b should be lowercase (f_b).
- The abbreviation "TDs" for Time-Domain Spectroscopy is incorrect and informal jargon. The commonly accepted abbreviation is "TDS" (Time-Domain Spectroscopy).
Despite authors claims about method capabilities such as "non-destructive and non-contact analysis, reveal internal structures, and detect invisible defects," etc., the conclusions of the paper, along with the general analysis and discussion, remain rather trivial for THz FMCW imaging and lack substantial novelty. The primary interest of this work lies in examining a specific object—an ancient enamel plate. However, the provided analysis appears superficial. It does not demonstrate deep intellectual effort in extracting unique information using FMCW. For example, although the authors claim the possibility of "pigment differentiation," this is not apparent from the provided data (especially considering the low quality of the images). It is unclear what advantages FMCW inspection provides compared to conventional visual inspection or other nondestructive testing (NDT) methods for this particular case, based on what is actually presented in the article. At the same time, significantly more advanced methods for FMCW data processing have been demonstrated, such as three-dimensional reconstruction of layered structures (e.g., in [A1]), which could provide deeper insight into the object’s structure. However, this work limits the analysis to depth images without detailed interpretation. Moreover, compared with previous research by the same authors on this enamel plate using TDS ([A2]), the previous analysis was considerably more informative. Thus, in its current form, the authors have failed to convincingly demonstrate the advantages or necessity of using FMCW specifically. A significant revision is required for publication.
[A1] Xue, K.; Zhang, W.; Song, J.; Wang, Z.; Jin, Y.; Nandi, A.K.; Chen, Y. Three-Dimensional Reconstruction Method for Layered Structures Based on a Frequency Modulated Continuous Wave Terahertz Radar. // Opt. Express, 2024, 32, 27303–27316, doi:10.1364/OE.528258.
[A2] Fauquet F. et al. Terahertz time-domain spectro-imaging and hyperspectral imagery to investigate a historical Longwy glazed ceramic // Sci Rep. Nature Publishing Group, 2024. Vol. 14, â„– 1. P. 19248.
Author Response
Review 3
The presented article, "Terahertz frequency modulated continuous wave inspection of an ancient enamel plate," is dedicated to the application of terahertz (THz) frequency-modulated continuous wave (FMCW) imaging for the nondestructive inspection of an ancient enamel plate. The topic is relevant both for the preservation of cultural heritage and, more broadly, for nondestructive testing. The paper compares systems operating at 100 GHz and 300 GHz in reflection and transmission modes. However, despite the relevance of the topic, the strong team of authors, and their recognized contributions to the field, the article in its current form contains several significant shortcomings:
- The manuscript exhibits serious inconsistencies in notations and definitions, causing confusion:
- Variables Z and d are both defined as "distance to the object." Are they synonyms, or do they represent different quantities? interchangeable terms for frequency bandwidth?
- Quantities dZ ("depth resolution") and δres ("longitudinal resolution") typically represent synonyms. However, from the formulas provided, the impression arises that these are different parameters. These terms must be clearly defined and consistently applied. Such carelessness is unacceptable for a scientific publication. - Are Δf ("bandwidth of the frequency modulation") and BW ("accessible bandwidth")
Response:
We thank the reviewer for pointing out this inconsistency. The variables Z and d were unintentionally used interchangeably in the original manuscript, which we recognize can cause confusion. In the revised version, we will adopt a consistent notation throughout the manuscript:
- We will use Z to denote the depth (or range) to the object along the radar line-of-sight.
- Δf has been removed to avoid confusion and is equal to BW
- δres is replaced with dZ
The manuscript includes inaccurately cited references. In the Introduction, it is claimed that the FMCW method enables inspecting samples with thicknesses of ~5–15 cm, citing references [11] and [12]. However, these references do not show the possibility of analyzing objects of such thickness. Reference [11] provides results for relatively thin items (scissors, CDs) and [12] for flat USAF test . Reference [15] describes research conducted with the TDS method rather than FMCW, as one might assume from the context.
Response:
References 11, 12, 15 have been changed.
The paper is prepared and formatted carelessly. Only some detected issues are listed below, as compiling a complete list would be overly exhaustive. Authors should be significantly more attentive to manuscript preparation:
- Presented THz scan images have low resolution. Axes and details in images are poorly readable, hindering a comprehensive assessment of results and complicating verification of the authors’ interpretations. This is a critical drawback for a paper focusing on the imaging.
- In lines 159-165, the font appears excessively condensed, making it unpleasant to read.
- The phrase in line 77 ("ethical approval, must list...") appears as incomplete placeholder text and needs correction or removal.
- In line 179, a period after reference [18] is missing.
- In the notation of beat frequency fb, the index b should be lowercase (f_b).
- The abbreviation "TDs" for Time-Domain Spectroscopy is incorrect and informal jargon. The commonly accepted abbreviation is "TDS" (Time-Domain Spectroscopy).
We sincerely thank the reviewer for this important feedback. We fully acknowledge that the initial submission contained formatting inconsistencies and lacked the level of precision expected in a scientific manuscript. In response, we have undertaken a comprehensive review and revision of the entire paper to address not only the issues pointed out by the reviewer but also many others that were identified during proofreading.
We have corrected inconsistencies in notation, improved the clarity and coherence of explanations, restructured sections where necessary, and ensured that formatting (including equations, figures, references, and variable definitions) adheres to professional and journal standards. We are confident that the revised manuscript reflects a much higher level of quality and rigor, and we greatly appreciate the reviewer’s constructive criticism, which has led to significant improvements in our work.
Despite authors claims about method capabilities such as "non-destructive and non-contact analysis, reveal internal structures, and detect invisible defects," etc., the conclusions of the paper, along with the general analysis and discussion, remain rather trivial for THz FMCW imaging and lack substantial novelty. The primary interest of this work lies in examining a specific object—an ancient enamel plate. However, the provided analysis appears superficial. It does not demonstrate deep intellectual effort in extracting unique information using FMCW. For example, although the authors claim the possibility of "pigment differentiation," this is not apparent from the provided data (especially considering the low quality of the images). It is unclear what advantages FMCW inspection provides compared to conventional visual inspection or other nondestructive testing (NDT) methods for this particular case, based on what is actually presented in the article. At the same time, significantly more advanced methods for FMCW data processing have been demonstrated, such as three-dimensional reconstruction of layered structures (e.g., in [A1]), which could provide deeper insight into the object’s structure. However, this work limits the analysis to depth images without detailed interpretation. Moreover, compared with previous research by the same authors on this enamel plate using TDS ([A2]), the previous analysis was considerably more informative. Thus, in its current form, the authors have failed to convincingly demonstrate the advantages or necessity of using FMCW specifically. A significant revision is required for publication.
[A1] Xue, K.; Zhang, W.; Song, J.; Wang, Z.; Jin, Y.; Nandi, A.K.; Chen, Y. Three-Dimensional Reconstruction Method for Layered Structures Based on a Frequency Modulated Continuous Wave Terahertz Radar. // Opt. Express, 2024, 32, 27303–27316, doi:10.1364/OE.528258.
[A2] Fauquet F. et al. Terahertz time-domain spectro-imaging and hyperspectral imagery to investigate a historical Longwy glazed ceramic // Sci Rep. Nature Publishing Group, 2024. Vol. 14, â„– 1. P. 19248.
We agree that the first version of the manuscript lacked sufficient depth in both analysis and interpretation, particularly in demonstrating the unique contributions of FMCW imaging as compared to visual inspection, other NDT methods, and our previous work using TDS ([A2]). In response, we have undertaken a substantial revision of the manuscript with the following improvements:
- Clearer Positioning of FMCW Advantages:
We now explicitly explain where FMCW imaging provides added value, especially in scenarios where TDS systems are impractical due to system complexity, acquisition time, or integration challenges. For example, FMCW enables faster scanning, greater compactness, and simplified acquisition, making it more suitable for in situ heritage diagnostics, which was a key motivation of our study. - More Detailed Image Analysis and Interpretation:
We have expanded the discussion about ring pattern and absorption impact. While pigment differentiation remains a challenge at this stage with a limited available bandwidth, we have revised our claims to avoid overstating this capability and clearly indicate it as a direction for future work, not a demonstrated outcome here. - Comparison to TDS and Related Work:
We now include a dedicated section comparing FMCW and TDS results on the same enamel plate, referencing [A2] directly. We explain that while TDS provides richer spectral information, FMCW offers practical advantages in deployment, and the current study focuses on validating its imaging potential in real-world heritage applications. - Discussion of Advanced FMCW Methods ([A1]):
We acknowledge the advanced techniques presented in [A1] and have cited this reference in our revised discussion. We clarify that our current goal was not to perform full 3D reconstruction but to establish baseline FMCW imaging capabilities for historical enamel artifacts. Nonetheless, we now outline a future plan to extend our analysis toward 3D structural reconstruction using similar approaches. - Revised Conclusion and Claims:
To reflect these improvements, the conclusion has been rephrased to better align with the actual findings, avoid overgeneralizations, and clearly state the scope and limitations of the study. We no longer make claims about pigment identification based on the current data, and instead focus on the feasibility and practical implications of applying FMCW imaging in cultural heritage.
We sincerely thank the reviewer for these valuable observations, which have led to a significant improvement in the manuscript. We believe that the revised version now provides a more rigorous, honest, and informative assessment of FMCW capabilities in the context of historical artifact analysis.
Round 2
Reviewer 2 Report
Comments and Suggestions for Authors
The paper by F. Fauquet, et al “Terahertz frequency modulated continuous wave inspection of an ancient enamel plate” has been modified and updated according to referees’ comments, all reported drawbacks were eliminated. The authors have made the manuscript more complete and comprehensive than the original version. Nevertheless, some points in the paper need clarification and improvement:
- Page 2, line 64 “Silicon radars “
Do you mean the company "Silicon Radar GmbH" or a semiconductor-based radar system? - Page 5, Figure 3a. This figure should be modified and enlarged - in its current form, neither the concentric patterns due to ceramic thickness variations nor the axis marks can be resolved.
The last request is valid for figure 3b. - Page 7, lines 244, 245. Please report parameters that were varied for data presented in Video 1; in particular distance between object and receiver (Z value).
- Page 7, Figure 4. This figure appears just at the beginnings of the section 2, before it was introduced in the text.
- There is well defined ring in Fig 4 that is no discussed in the text.
- Page 8, lines 286, 287. “The limited dynamic range”. In fact, the dynamic range limits in Fig. 4 are the same as in Fig. 3.
- Page 8, lines 299, 300. Please explain what is meant by "central darkest circular zone"
- There are a few questions concerning Fig 5, Page 8, lines 300 - 310:
- a) It is not very clear what this figure is supposed to show and why it is included in the article. After all, it compares the visible image and the TDS results; it would be more natural if the results of the FMCW and TDS were compared.
b) “image of the central zone of the plate”. In fact, the entire plate is shown.
c) In any case, the description of Fig. 5 (if any) should begin with a new paragraph.
Author Response
Reviewer 2
The paper by F. Fauquet, et al “Terahertz frequency modulated continuous wave inspection of an ancient enamel plate” has been modified and updated according to referees’ comments, all reported drawbacks were eliminated. The authors have made the manuscript more complete and comprehensive than the original version. Nevertheless, some points in the paper need clarification and improvement:
- Page 2, line 64 “Silicon radars “
Do you mean the company "Silicon Radar GmbH" or a semiconductor-based radar system?
We appreciate the reviewer’s comment. The sentence is not clear , we are thinking about silicon based integrated radars such “ Silicon Radar GmbH , or 2PI-LAbs GmbH “
- Page 5, Figure 3a. This figure should be modified and enlarged - in its current form, neither the concentric patterns due to ceramic thickness variations nor the axis marks can be resolved.
The last request is valid for figure 3b.
Thank you for this remark. We enlarged the picture inside the doc file.
- Page 7, lines 244, 245. Please report parameters that were varied for data presented in Video 1; in particular distance between object and receiver (Z value).
We appreciate the reviewer’s comment, we add :
For this experiment, the plate is positioned at 50mm of a Teflon lens.
- Page 7, Figure 4. This figure appears just at the beginnings of the section 2, before it was introduced in the text.
We rearranged the text and enhanced the figures
- There is well defined ring in Fig 4 that is no discussed in the text.
- Page 8, lines 299, 300. Please explain what is meant by "central darkest circular zone"
- There are a few questions concerning Fig 5, Page 8, lines 300 - 310:
- a) It is not very clear what this figure is supposed to show and why it is included in the article. After all, it compares the visible image and the TDS results; it would be more natural if the results of the FMCW and TDS were compared.
b) “image of the central zone of the plate”. In fact, the entire plate is shown.
c) In any case, the description of Fig. 5 (if any) should begin with a new paragraph.
We appreciate the reviewer’s comment. All these interrogations are related to the same thing.
This description is not clear .
So, a typical plate consists of several distinct parts: The well is the central, recessed area where food is served. The lip surrounds the well and is usually a flat or slightly inclined section that rises gently or runs parallel to the base. While often mistaken for the rim, the lip varies in width depending on the design and is not always present. The rim refers to the outermost edge of the plate, which is often decorated—for example, with gilding or painted details. The base is the underside of the plate, sometimes featuring a foot ring to elevate it slightly from the surface it rests on. What we chiefly detect is the base like with X-Ray
Updated figure 5
So we rewrite the paragraph
At least, a central darkest circular zone is visible for both frequencies. We also provided in Fig. 5a, the superposition of the visible and a X-Ray images of the central part of the plate and Fig. 5b rear visible image of the central zone of the plate, combined with a THz imaging scan performed with a time domain spectro imaging setup in reflection. So, a typical plate consists of several distinct parts. The well is the central, recessed area where food is served. The lip surrounds the well and is usually a flat or slightly inclined section that rises gently or runs parallel to the base. While often mistaken for the rim, the lip varies in width depending on the design and is not always present. The rim refers to the outermost edge of the plate, which is often decorated—for example, with gilding or painted details. The base (shown by a red dot circle in Fig. 5) is the underside of the plate, sometimes featuring a foot ring to elevate it slightly from the surface it rests on. In transmission mode, this added thickness of the base hides and alters the analysis in the central part of the plate of the surface details (See Fig 4 for FMCW transmission images and Fig. 5a for X-Ray image). The TDS THz image (inserted in the lower part) reveals internal structures of the plate that are invisible to the naked eye nor on FMCW images. This is probably linked to variations in reflected field and, indicated differences in density during the firing of the ceramic. and also, during the casting of the piece. Plaster molds are generally used for the manufacture of pieces.
- Page 8, lines 286, 287. “The limited dynamic range”. In fact, the dynamic range limits in Fig. 4 are the same as in Fig. 3.
Thank you for this remark. We rewrite the text as follow:
The dynamic range observed in the transmission images (Fig. 4) is comparable to that in the reflection images (Fig. 3). However, the smoother contrast in transmission mode is primarily due to different propagation mechanisms and attenuation effects, rather than a limitation in dynamic range.
Reviewer 3 Report
Comments and Suggestions for Authors
I have read the revised manuscript and the authors’ response to my earlier comments. I appreciate their efforts to address some of the points I raised, especially regarding notation problems and citation changes. I also recognize that the authors have tried to make some claims less strong, such as those about pigment differentiation, and have acknowledged the advanced techniques used in [22].
However, even with these changes, several important issues from my previous review are still not fully resolved. Also, the new version of the manuscript brings up some new concerns. I believe the manuscript still needs major revisions before it can be accepted for publication.
- The revised text says in line 255: "Cross-sectional depth profiles have been included to reveal variations in enamel thickness and potential subsurface inhomogeneities." However, these profiles are absent from the manuscript figures. As these profiles are presented in the text as key evidence derived from the expanded analysis, their omission is a major flaw.
- The THz images still have quality and clarity problems. They are not good enough for a paper that focuses on imaging. For example, in Figure 3a, a large part of the color scale (from -30 dB to -50 dB) looks saturated (white), which hides possible details and makes it hard to see the ring structure the authors talk about. Also, the color bars in Figures 3a and 3b are opposite—one shows high intensity as dark, the other as light. This makes it confusing for the reader to compare the 100 GHz and 300 GHz results. Figure 3c is also missing a color scale and coordinate axes.
- There is a mismatch between the experimental setup described in Section 3 and the block diagram in Figure 2. The text mentions a VCO, DAQ, and DDS, but these are not labeled in Figure 2. Instead, Figure 2 shows a "Ramp generator" and "ADC." These might be related, but the names must be the same in the text and figure. Also, the PA component is not labeled in the figure.
- While the authors have toned down the claim of "pigment differentiation," the discussion still suggests a correlation between reflectivity variations at 300 GHz and specific pigments (pink, green), heavily relying on prior chemical validation in [18]. However, this is not clearly shown in the images (Figures 3 and 4). I suggest to highlight specific regions in the THz images and explicitly link them to the corresponding colored areas (e.g., using arrows or outlines correlated with a visible light image legend). Without this kind of clear visual support, even the weaker claim of "correlation" is not well supported and apparent from the data.
- In the revised text the authors claims compactness as a key advantage of FMCW systems, particularly for in situ work. But they do not give any information about the dimension and weight of their 100/300 GHz system. They also do not compare it to other systems. Without comparing its size/weight/setup complexity to typical TDS systems or other NDT tools, the claim of "compactness" is poorly substantiated.
In summary, I appreciate the authors’ efforts, but the manuscript still has serious issues. These issues make it hard to properly evaluate the value of the work and the claimed benefits of the FMCW method. I recommend another round of major revisions. The authors should include the missing data, improve the figures, and make sure the description is clear and consistent.
Author Response
Reviewer 3
I have read the revised manuscript and the authors’ response to my earlier comments. I appreciate their efforts to address some of the points I raised, especially regarding notation problems and citation changes. I also recognize that the authors have tried to make some claims less strong, such as those about pigment differentiation, and have acknowledged the advanced techniques used in [22].
However, even with these changes, several important issues from my previous review are still not fully resolved. Also, the new version of the manuscript brings up some new concerns. I believe the manuscript still needs major revisions before it can be accepted for publication.
- The revised text says in line 255: "Cross-sectional depth profiles have been included to reveal variations in enamel thickness and potential subsurface inhomogeneities." However, these profiles are absent from the manuscript figures. As these profiles are presented in the text as key evidence derived from the expanded analysis, their omission is a major flaw.
We agree that the sentence on line 255 — "Cross-sectional depth profiles have been included to reveal variations in enamel thickness and potential subsurface inhomogeneities" — was not sufficiently precise. We acknowledge that no such profiles are shown in the current set of figures, and more importantly, that the depth resolution of our FMCW radar system (~2 mm) is not sufficient to allow accurate extraction of enamel layer thickness, which is typically below 1 mm.
To address this, we have revised the text to clarify that while variations in reflected signal intensity and interference patterns suggest changes in subsurface structure, these observations do not constitute resolved depth profiles in the conventional sense. Instead, they provide indirect indications of structural inhomogeneities, including possible enamel thickness variation or density gradients, inferred from contrast and spatial signal modulations.
We appreciate the reviewer’s insight and have updated the manuscript accordingly to avoid any misinterpretation.
- The THz images still have quality and clarity problems. They are not good enough for a paper that focuses on imaging. For example, in Figure 3a, a large part of the color scale (from -30 dB to -50 dB) looks saturated (white), which hides possible details and makes it hard to see the ring structure the authors talk about. Also, the color bars in Figures 3a and 3b are opposite—one shows high intensity as dark, the other as light. This makes it confusing for the reader to compare the 100 GHz and 300 GHz results. Figure 3c is also missing a color scale and coordinate axes.
Figure 3. Reflected FMCW THz images of the enamel plate acquired at two frequencies:
(a) 100 GHz image highlighting large-scale interference patterns and structural features, with improved dynamic range (–35 dB to –50 dB) to enhance contrast without saturation;
(b) 300 GHz image revealing finer surface textures and localized heterogeneities, using the same color scale convention as in (a) to facilitate comparison;
(c) zoomed-in view of the central motif (phoenix) in the 300 GHz image, presented with color scale and coordinate axes for accurate spatial and intensity interpretation.
Updated Fig 3
- There is a mismatch between the experimental setup described in Section 3 and the block diagram in Figure 2. The text mentions a VCO, DAQ, and DDS, but these are not labeled in Figure 2. Instead, Figure 2 shows a "Ramp generator" and "ADC." These might be related, but the names must be the same in the text and figure. Also, the PA component is not labeled in the figure.
We thank the reviewer for identifying the mismatch between the terminology used in the description of the experimental setup (Section 3) and the labelling in Figure 2.
We fully agree that consistency between the figure and the main text is essential for clarity and technical accuracy, especially in a paper dealing with signal chain components.
To address this:
- Terminology alignment: We have revised the text in Section 3 and the caption of Figure 2 to ensure consistent terminology throughout. Specifically, the component referred to as the “Ramp generator” in Figure 2 corresponds functionally to the Direct Digital Synthesizer (DDS) mentioned in the text. This has been clarified both in the figure and the associated description. The “ADC” in Figure 2 performs the role of the Data Acquisition (DAQ) system mentioned in the text. We have added a clarifying note to the figure caption. The Voltage-Controlled Oscillator (VCO) and its phase-locked loop (PLL) function were previously described in the text but were not explicitly labeled in the diagram. We have now added these components and indicated their role in frequency linearization. Missing PA (Power Amplifier) label:
The Power Amplifier (PA), which boosts the modulated signal before transmission, was unintentionally omitted from the diagram. We have now added and labeled this component in the updated version of Figure 2. Updated Figure 2:
We have revised Figure 2 to clearly include and label the VCO, DDS, DAQ (ADC), PLL, and PA components. Arrows and block connections have been double-checked for accuracy and consistency with the described signal flow.
We believe these updates significantly improve the clarity and technical coherence of the manuscript, and we are grateful to the reviewer for helping us address this important detail.
Updated figure 2
- While the authors have toned down the claim of "pigment differentiation," the discussion still suggests a correlation between reflectivity variations at 300 GHz and specific pigments (pink, green), heavily relying on prior chemical validation in [18]. However, this is not clearly shown in the images (Figures 3 and 4). I suggest to highlight specific regions in the THz images and explicitly link them to the corresponding colored areas (e.g., using arrows or outlines correlated with a visible light image legend). Without this kind of clear visual support, even the weaker claim of "correlation" is not well supported and apparent from the data.
We appreciate the reviewer’s thoughtful observation and agree that a clear visual connection between specific pigment regions and reflectivity variations in the THz images would strengthen the discussion.
While we have intentionally moderated our claim from "pigment differentiation" to "possible correlation," we acknowledge that this interpretation relies in part on prior chemical analyses presented in our earlier study [18]. However, we understand that the current figures (particularly Figures 3) do not visually convey this correlation in an explicit or self-evident manner.
To address this, we have taken the following actions:
Visual annotations added to Figures 3c:
We have updated these figures to include circles identifying regions of interest where notable reflectivity variations were observed—specifically in areas corresponding to pink, red and dark green pigments, such as the phoenix’s throat and chest, and foliage motifs. These markings are now linked to the visible light reference image (Figure 5a), which serves as a guide to the decorative motifs and their colors.
Clarification in the discussion:
In Section 5 (Discussion), we have added a sentence to clarify that although these reflectivity patterns are consistent with pigment locations, the THz imaging itself cannot unambiguously confirm chemical composition. Instead, the THz contrast serves as a complementary indicator, likely influenced by both dielectric properties and surface geometry.
We hope these improvements provide the clarity needed to support the more cautious interpretation of pigment-related reflectivity variations
- In the revised text the authors claims compactness as a key advantage of FMCW systems, particularly for in situ work. But they do not give any information about the dimension and weight of their 100/300 GHz system. They also do not compare it to other systems. Without comparing its size/weight/setup complexity to typical TDS systems or other NDT tools, the claim of "compactness" is poorly substantiated.
We appreciate the reviewer’s suggestion to provide quantitative details and comparative context to substantiate the claim of “compactness” for the FMCW system.
In response, we have added the following clarifications to the manuscript:
- System dimensions and weight:
we now specify the approximate size and weight of our transceiver modules:
“The 100/300 GHz FMCW imaging heads used in this study are integrated into compact units measuring approximately 22 × 18 × 12 cm³ and weighing around 2.5 kg each (excluding the laptop-based data acquisition system).”
- Comparison with TDS systems:
In the Discussion section, we now explicitly contrast our FMCW setup with typical time-domain spectroscopy (TDS) systems:
We add : Compared to conventional time-domain terahertz spectroscopy (TDS) systems, which typically involve femtosecond lasers, optical delay stages, and bulky benchtop configurations, the FMCW system could be more compact and field-adaptable. With transceiver modules measuring approximately 120 mm x 60 mm x 70 mm (L/H/W) mm³ and weighing less a kg [23], the setup is easily portable. These features allow for flexible deployment in non-laboratory environments such as museums, conservation sites, or archaeological contexts.
To highlight the practical impact of this advantage, we added a statement to the Conclusion emphasizing the system’s portability and relevance for conservation and archaeological diagnostics.
We believe these additions provide a clear and evidence-based justification for describing FMCW radar as “compact,” especially in the context of mobile, non-invasive heritage analysis.
In summary, I appreciate the authors’ efforts, but the manuscript still has serious issues. These issues make it hard to properly evaluate the value of the work and the claimed benefits of the FMCW method. I recommend another round of major revisions. The authors should include the missing data, improve the figures, and make sure the description is clear and consistent.
Round 3
Reviewer 3 Report
Comments and Suggestions for Authors
Dear authors,
Unfortunately, the revised manuscript demonstrates insufficient care in addressing the previously raised comments.
In Figure 2, the explanation of the abbreviation "LNA" (Low Noise Amplifier), which was present in the earlier caption, has now disappeared, though the abbreviation itself remains in the figure. Additionally, the new Figure 3(c) raises confusion: the coordinate axes are labeled as "50 mm" (x-axis) and "100 mm" (y-axis), which is difficult to interpret—does this imply the image scale is 1:2? Furthermore, the rightmost circular elements differ noticeably between the THz and visible images: the circles are touching in the THz image but not in the visible image. What is the reason for this discrepancy?
Most importantly, the correlation claimed by the authors between color pigments and THz contrast lacks quantitative support and appears unconvincing. At present, this represents the most serious assumption made by the authors and should either be clearly demonstrated with quantitative data or omitted altogether.
Finally, the system dimensions and weight provided in the manuscript (120 x 60 x 70 mm3, <1 kg) contradict those mentioned in your response (22 x 18 x 12 cm3, 2.5 kg), further underscoring the manuscript’s lack of thoroughness.
These issues must be carefully resolved before the paper can be considered for publication.
Author Response
Unfortunately, the revised manuscript demonstrates insufficient care in addressing the previously raised comments.
In Figure 2, the explanation of the abbreviation "LNA" (Low Noise Amplifier), which was present in the earlier caption, has now disappeared, though the abbreviation itself remains in the figure.
We sincerely thank the reviewer for highlighting this oversight. We have reintroduced the full definition of "LNA" (Low Noise Amplifier) in the caption of Figure 2 to ensure clarity for all readers, including those less familiar with high-frequency radar terminology. The caption now fully defines all abbreviations used in the figure.
The reflected signal, delayed due to the propagation back and forth to the object, is first amplified by a Low Noise Amplifier (LNA) to minimize noise, then guided toward the detection circuit with a directional coupler linked to a multiplexer where it is mixed with the reference signal.
Additionally, the new Figure 3(c) raises confusion: the coordinate axes are labeled as "50 mm" (x-axis) and "100 mm" (y-axis), which is difficult to interpret—does this imply the image scale is 1:2? Furthermore, the rightmost circular elements differ noticeably between the THz and visible images: the circles are touching in the THz image but not in the visible image. What is the reason for this discrepancy?
We thank the reviewer for this relevant question. Based on the scale provided along the axes in Figure 3(a), we have estimated the size of the central bright region—corresponding to the main decorative motif of the plate—directly from the image data.
Along the x-axis, the high-reflectivity area extends approximately from 230 mm to 425 mm, giving a lateral dimension of about 200 mm. Along the y-axis, a vertical dimension of about 240 mm is found. These values are now mentioned in the revised manuscript to provide a better spatial reference for readers when interpreting the radar image.
We also modified the scale information to report the real dimensions on the zoomed figure
We add : The central bright region in Fig.3a and zoomed in Fig .3c, spans approximately 200mm along the x-axis and 240 mm along the y-axis, based on the figure’s axis scale.
New figure 3
Most importantly, the correlation claimed by the authors between color pigments and THz contrast lacks quantitative support and appears unconvincing. At present, this represents the most serious assumption made by the authors and should either be clearly demonstrated with quantitative data or omitted altogether.
We thank the reviewer for raising this essential point. We fully agree that the correlation between THz reflectivity and pigment composition should be treated with caution in the absence of quantitative chemical analysis directly linked to the THz maps.
In the current version of the manuscript, the statements about pigment-contrast correlation were based on clear visual evidence from the 300 GHz reflectivity maps: the boundaries of certain colored regions—such as pink, green, and golden enamel areas—are distinctly outlined in the THz images, even though no direct pigment identification was attempted via THz alone. These contours correspond spatially to known decorative motifs, as verified from the visible reference image (Fig. 1), and appear consistently in both reflection and transmission modes.
That said, we fully acknowledge that this does not constitute a validated pigment-specific mapping. As such, we have:
Rephrased the relevant parts of the manuscript to avoid implying any strong claim of pigment identification;
Clearly stated that the observed contrast likely reflects a combination of surface morphology and dielectric property variations, which may or may not be linked to pigment content;
Noted that previous studies (e.g., Ref. [18]) using XRF and VNIR have confirmed elemental differences in these regions, but no pixel-level registration with the THz maps has been performed in this work;
Added a recommendation in the Discussion that future work should involve co-registered imaging (THz + hyperspectral or XRF) to rigorously investigate this correlation quantitatively.
We hope that this revised, more cautious formulation respects the reviewer’s concern while preserving the relevance of the observed spatial contrasts, which remain meaningful in a structural and conservation context.
New text : The reflectivity maps at 300 GHz clearly reveal well-defined outlines that coincide with some of the colored motifs visible in the optical image, particularly areas painted with pink, green, and gold enamels. These spatially distinct zones suggest that terahertz imaging is sensitive to surface morphology and dielectric property variations.
However, while these contrasts visually correspond to pigmented regions, they do not constitute direct evidence of pigment differentiation. No quantitative chemical analysis or pixel-level co-registration was performed in this study to formally link the THz reflectivity variations to material composition.
Previous analyses using XRF and VNIR hyperspectral imaging (Ref. [18]) confirmed that certain areas are enriched with elements such as Cu, Co, Cr, or Mn, but without synchronized comparison with the THz maps.
Consequently, the observed contrasts are considered suggestive but not conclusive of dielectric variations related to pigments. Future work should combine co-registered THz imaging with chemical mapping techniques to rigorously assess the correlation between pigment composition and terahertz reflectivity.
Finally, the system dimensions and weight provided in the manuscript (120 x 60 x 70 mm3, <1 kg) contradict those mentioned in your response (22 x 18 x 12 cm3, 2.5 kg), further underscoring the manuscript’s lack of thoroughness.
These issues must be carefully resolved before the paper can be considered for publication.
We sincerely thank the reviewer for carefully identifying this important inconsistency.
After thorough verification, we confirm that there was a confusion between the dimensions and weight of two different hardware components associated with the experimental setup:
The dimensions 120 mm × 60 mm × 70 mm and weight <1 kg correspond to the FMCW radar transceiver head only (i.e., the emitter/receiver block excluding the additional mechanical supports and electronics).
The dimensions 22 cm × 18 cm × 12 cm and weight ~2.5 kg refer to the full assembled FMCW imaging unit, including the transceiver head, mechanical scanning frame, supporting optics (Teflon lenses), and associated cables/mounts.
To avoid further confusion, the description now specifies that:
The compact dimensions (<1 kg) concern the radar transceiver itself;
We apologize for the lack of precision only present in the response letter.
